# Habitual Physical Activity and Diabetes Control in Young and Older Adults with Type II Diabetes: A Longitudinal Correlational Study

**DOI:** 10.3390/ijerph18031330

**Published:** 2021-02-02

**Authors:** Chia-Hsun Chang, Ching-Pyng Kuo, Chien-Ning Huang, Shiow-Li Hwang, Wen-Chun Liao, Meng-Chih Lee

**Affiliations:** 1Department of Nursing, Taichung Veterans General Hospital, Taichung 40705, Taiwan; up2u5400@vghtc.gov.tw; 2School of Nursing, Chung Shan Medical University, Taichung 40201, Taiwan; pyng@csmu.edu.tw; 3Department of Nursing, Chung Shan Medical University Hospital, Taichung 40201, Taiwan; 4Institute of Medicine, Chung Shan Medical University, Taichung 40201, Taiwan; cshy049@csh.org.tw; 5Department of Internal Medicine, Chung Shan Medical University Hospital, Taichung 40201, Taiwan; 6School of Nursing, Asia University, Taichung 41354, Taiwan; slhwang@asia.edu.tw; 7School of Nursing, China Medical University, Taichung 406040, Taiwan; 8Department of Nursing, China Medical University Hospital, Taichung 40447, Taiwan; 9Department of Family Medicine, Taichung Hospital, Ministry of Health and Welfare, Taichung 40343, Taiwan; 10Institute of Population Health Sciences, National Health Research Institutes, Miaoli 35053, Taiwan; 11College of Management, Chaoyang University of Technology, Taichung 413310, Taiwan

**Keywords:** daily physical activity, diabetes control, International Physical Activity Questionnaire (IPAQ), Type II Diabetes Mellitus (T2DM)

## Abstract

This study aimed to determine whether daily physical activity in young and older adults with T2DM is associated with diabetes control. A prospective correlational study involving 206 young (≤65 years) and older (>65 years) adults was conducted. The International Physical Activity Questionnaire was used to assess their daily physical activity levels. Patients’ mean HbA1c level was 7.8% (±1.4), and 95.9% of patients had unsatisfactory diabetes control. Performing more minutes per week of moderate-intensity daily physical activity was associated with a lower risk of glycemia in both young and older adults. Furthermore, moderate daily physical activity significantly lowered the risk of glycemia. Health personnel must encourage patients to engage in moderate daily physical activities to improve diabetes control.

## 1. Introduction

The prevalence of Type II Diabetes Mellitus (T2DM) is increasing worldwide and in Taiwan [1]. The increasing prevalence and incidence of T2DM inflict a considerable burden on healthcare systems due to long-term complications [1]. Age has an impact on mortality and vascular complications with T2DM, and young age at diabetes is associated with a 3~5% increased risk of mortality and complications [2]. Managing T2DM is crucial for preventing or controlling diabetes-related complications and even more important for younger diabetes [2]. The American Diabetes Association 2018 [3] suggested that T2DM should be controlled by maintaining the “ABC” standard, which refers to a patient’s hemoglobin A1c (HbA1c) levels, blood pressure, and low-density lipoprotein cholesterol (LDL-C) levels. Exercise can improve glucoregulatory activity, stimulate endothelial cells to release vasodilators (e.g., nitric oxide and bradykinin), increase baroreflex sensitivity, control blood pressure [4], facilitate the reverse transport of cholesterol, and accelerate mature rates of high-density lipoprotein [5]. Numerous studies have shown that exercise undertaken by patients with T2DM facilitates disease control [6,7,8,9,10]. The results of a meta-analysis of progressive resistance exercise showed that patients with T2DM who engaged in exercise three times a week for more than 8 weeks reduced their HbA1c levels by 0.3% [6]. Umpierre et al. (2011) further synthesized related literature and found that engaging in structured supervised exercise, including aerobic, resistance, or combined training exercises, three times a week for 150 min a week for 12 weeks or longer, reduced patients’ HbA1c levels by 0.67% [7]; however, undertaking exercise advice partially or participating in exercise that was not supervised, reduced HbA1c levels by 0.43% [7]. These results suggest the effectiveness of exercise, with varied intensity, patterns, and methods, for diabetes control.

Physical activity is defined as bodily movement produced by skeletal muscles with energy expenditure [11]. A simple category for the total energy expenditure produced by physical activity can be identified as at work and leisure, except for a small amount of energy production during sleep [11]. Thus, physical activity includes occupational and leisure-time activities, such as sports, conditioning, household chores, or other activities [11]. Exercise is a subcategory of physical activity and a part of physical activities, with total daily physical activities consisting of exercise and non-exercise activities. Performing physical activities, including non-exercise activities, may also affect the control of diabetes.

Physical activity is further divided into low-, moderate-, or heavy-intensity activities, which are mutually exclusive [11]. A study compared continuous low- to moderate-intensity endurance-type exercise (50% peak oxygen uptake, VO_2_ peak) and moderate- to high-intensity endurance-type exercise (75% VO_2_ peak) to investigate the intensity of physical activity required for diabetes control and found similar effects of both types on reducing HbA1c levels or LDL-C levels in obese T2DM patients when their medication and diet were well-controlled [8]. Another study compared different intensities of physical activity patterns and found similar effects of engaging in moderate exercise (60% maximal oxygen uptake, VO_2_ max) for 30 min every day or 60 min every other day on glycemic levels in patients with T2DM [9]. These findings suggest the positive effect of varied intensity and frequency in performing physical activities on glycemic control in diabetes. Moderate-intensity physical activity performed either daily or every other day is effective for glycemic control. However, even if the amount of these effective physical activities were counted by levels of intensity, these studies all focused on structured exercise, and the non-exercise part of physical activities was not counted. A total of 53.8% of Asians participated in low-intensity exercise, 35% in moderate physical activity, and 11.2% in high physical activity [12]. If non-exercise physical activity can be included, patients might be able to more readily achieve a sufficient amount of physical activity to control diabetes. 

A study compared the status of physical activities in patients whose diabetes was under control (HbA1c ≤ 7%, *n* = 93) with those whose diabetes was not under control (HbA1c > 7%, *n* = 117) and found that patients under control walked more minutes per week than did patients not under control (215.9 vs. 150.7 min/week, t = 2.49, *p* < 0.05) [10]. A low-intensity activity, such as walking 37 min per day for 6 days a week, was able to effectively control patients’ glycemic levels [10]. People perform both exercise and non-exercise physical activities, and if non-exercise physical activities such as household or work tasks are counted in addition to exercise, a positive effect of total physical activity on glycemic control may be seen for patients with diabetes.

Whether or not daily physical activities, including work, walking, housework, gardening, leisure activities, and transportation, have a similar effect on diabetes control is not well-studied. Mynarski et al. (2012) investigated the effect of habitual physical activities on unemployed or retired patients with T2DM by using an accelerometer and the International Physical Activity Questionnaire (IPAQ) and found that daily habitual physical activity did not influence HbA1c levels [13]; however, this study had a small sample size (*n* = 31), and a larger population is needed for similar future studies. Moreover, engaging in structured exercises may require supervision. Time for exercise is even limited in younger adults when their time is occupied with work. Comparatively, daily physical activities such as work, walking, housework, or leisure activities can be undertaken in everyday life and are therefore simple, convenient, and easy to practice. Since the IPAQ is correlated with accelerometer measurements and can be an easy tool for physical activity assessment [13] in a larger sample size, this study used the IPAQ to assess patients’ daily physical activity levels and explored whether levels of habitual physical activity undertaken by Taiwanese patients with T2DM were compatible with low-to-moderate-intensity exercise to generate benefits. Moreover, the age group divided by 65 years was stratified in this study to investigate if age plays roles on physical activity and glycemic control in type 2 diabetics.

## 2. Methods

### 2.1. Research Design

A cross-sectional survey with a 1-year medical follow-up was adopted in this study. Data were collected through structured interviews and medical record reviews at the endocrinology outpatient clinic of a medical center in Central Taiwan. Physicians in endocrine clinics asked for permission from diabetes patients and referred them to this study. Self-reported habitual physical activity was investigated through face-to-face interviews. Diabetes control, including HbA1c, blood pressure, and LDL-C levels, were followed-up every 3 months for 1 year after recruitment.

### 2.2. Participants

Diabetes patients aged 30 to 85 years, diagnosed with T2DM, who had no history of mental disorders (e.g., schizophrenia) or cognitive impairment (Mini-Mental State Examination, MMSE > 25 in those with a middle to high educational level, or 19 in those with a low education level or who were illiterate), could speak Mandarin Chinese or Taiwanese (or their family members could speak one of the languages), and lived in the community, were recruited.

Patients who were diagnosed with type I diabetes, gestational diabetes, exocrine pancreatic disease, drug-induced diabetes, or severe cardiovascular disease were excluded from this study. In addition, those who had severe depression or anxiety (Hospital Anxiety and Depression Scale, HADS > 5), cognitive impairment (MMSE ≤ 19), a history of alcohol or drug abuse, or who were hospitalized during follow-up, were excluded. Two-hundred-and-seventy-six patients were approached, of which 5 refused to participate and 65 were excluded from the analysis because of incomplete laboratory data. Two-hundred-and-six patients completed the study. Figure 1 shows the study recruitment procedure. This study was approved by the Institutional Review Board of Chun Shan Medical University Hospital (CSMUH CS09025). All the participants gave their informed consent to participate in this study.

### 2.3. Measurements

Demographic information. Personal data, including age, sex, occupation, work conditions, education level, marital status, living conditions, and number of years diagnosed with diabetes, were collected through face-to-face interviews. Patient privacy was respected during the interview.

Assessment of habitual physical activity. A short version of the IPAQ Taiwan version [14] was used in this study to assess patients’ physical activity levels. The validity of the Content Validity Index of the IPAQ Taiwan version was greater than 0.8, and the consistency between the English and Taiwan versions was greater than 0.9, with a reliability of the intraclass correlation coefficient of 0.7–0.9 [14]. This questionnaire was employed to determine the average amount of time that the participants spent walking and performing high- and moderate-intensity activities during a 7-day period and the average amount of time spent sitting during a workday [14]. Only physical activity that continued for a minimum of 10 min was counted. Physical activity levels were estimated using a metabolic equivalent task (MET): (1) vigorous- or high-intensity activity refers to an activity of greater than 6 METs, such as running and climbing; (2) moderate-intensity activity refers to an activity of 3 to 6 METs, such as swimming and dancing; and (3) low-intensity activity refers to an activity of <3 METs, such as walking. The formula for calculating the amount of total physical activity per week (MET-min/wk) is as follows: 3 × walking (min) × number of days + 4.5 × moderate-intensity activity (min) × number of days + 6 × high-intensity activity (min) × number of days [14]. Three types of lifestyles were derived based on the total MET per week: (1) a high-intensity lifestyle denoted a person who undertook high-intensity activities for a minimum of 3 days per week, achieving a total physical activity level of 1500 MET-min/wk or a patient whose total physical activity (including walking and moderate- or high-intensity activities) for 7 days per week reached 3000 MET-min/wk; (2) a moderate-intensity lifestyle was defined as a person who engaged in high-intensity activities 20 min per day for a minimum of 3 days per week, undertook moderate-intensity activities for 30 min per day for a minimum of 5 days per week, walked 30 min per day for 7 days per week or who walked and engaged in moderate- or high-intensity activities for >5 days per week, with a total physical activity level reaching 600 MET-min/wk; and (3) a low-intensity lifestyle denoted a person whose physical activity level did not meet any of the aforementioned criteria (International Physical Activity Questionnaire, n. d.).

Diabetes control. According to the ADA and guidelines in Taiwan [15], diabetes control is defined as HbA1c ≤ 7%, blood pressure < 140/90 mmHg in older (>65 years) adults and as <130/80 mmHg in younger adults, and LDL-C < 100 mg/dL. After recruitment, the laboratory data of diabetes control in each participant was followed every 3 months for 1 year. These data were obtained from medical records, prospectively, after study enrollment.

### 2.4. Data Analysis

Descriptive statistics, including frequency, percentage, mean, and standard deviation, were used to understand sample distributions of physical activity level and control of diabetes. The Pearson correlation test was used to examine the univariate correlation between habitual physical activity and control of diabetes. Multiple logistic regression was used to examine the association between habitual physical activity and control of diabetes by using significant factors from the Pearson correlation test. Age, sex, and number of years diagnosed were set as confounding factors in the regression model. The significance level was set at *p* < 0.05 and was two-tailed. The Statistical Package for Social Sciences (SPSS) for Windows, Version 26 software (SPSS Inc., Chicago, IL, USA) was used to analyze the data.

## 3. Results

### 3.1. Demographic Characteristics

A total of 271 patients participated in this study, 65 of whom withdrew with a completion rate of 76%. Table 1 shows the demographic characteristics and diabetes control status of the participants. In this study, the average age of the participants was 62.5 years (±10.4 years); more than half of the patients were women, and the average duration of diabetes since diagnosis was 13.2 years (±7.8 years). Since the patients were distributed across a wide age range (33~82 years) and their physical health, activity levels, and comorbidities might have varied with age, the patients were divided into younger (≤65 years) and older (>65 years) groups. In the younger group, more patients were working, had a higher education, and were more obese than those in the older group (Table 1).

### 3.2. Diabetes Control in Young and Older Patients with T2DM

Table 1 presents the status of diabetes control. For both younger and older patients with diabetes, the mean HbA1c level was 7.8% (±1.4). Only one-third of participants (30.1%) had good glycemic control (HbA1c ≤ 7%), less than half (36.9%) had appropriate blood pressure, and less than half (44.2%) had appropriate lipid control. Only 8 patients (4.1%) presented with normal diabetes control. Younger patients had poorer diabetes control in terms of blood pressure and LDL-C than older patients (Table 1).

### 3.3. Daily Habitual Physical Activity in Young and Older Patients with T2DM

The IPAQ was used to assess patients’ daily habitual physical activity, as shown in Table 2. In a week, patients spent 4 days participating in low-intensity activities, with 43.8 min per day on average, and only spent 1.6 days a week with 17.3 min a day participating in moderate- to high-intensity activities. The mean total physical activity for older patients was 1408.9 METs/week and 1373.4 METs/week for younger patients. Physical activity was categorized into three types of dynamic activity. The majority of both younger and older patients engaged in low- to moderate-intensity activities and only one-tenth (11.1%) engaged in high-intensity activities. The average time spent in sedentary behaviors was about 4.8 h and 5.2 h a day for younger and older patients, respectively. The younger patients participated in higher intensity activity but less moderate-intensity activity per week than did the older patients. There were no differences in other activity levels between the younger and older groups (Table 2).

### 3.4. Association of Habitual Physical Activity with Diabetes Control in Patients with T2DM

The Pearson correlation test was performed to examine the association between habitual physical activity and diabetes control (Table 3). The results in all patients showed that performing moderate-intensity activities multiple days a week and minutes a day was significantly correlated with normal levels of blood sugar. Moreover, performing high-intensity activities multiple days a week was correlated with higher levels of blood pressure. However, when the analysis was separated for the younger and older groups, only days a week in performing moderate-intensity activity was significantly correlated with normal blood sugar levels in younger patients.

Logistic regression analysis was used to analyze the association between habitual physical activity levels and diabetes control in patients with T2DM based on the results of the above Pearson correlation. As shown in Table 4, the duration of moderate-intensity activity for minutes per day was significantly associated with normal glycemic control, after adjusting for age, sex, and number of years diagnosed with T2DM (Table 4). Every minute per day increased in performing moderate-intensity activity lowered the risk of poor glycemic control by 0.1%.

## 4. Discussion

### 4.1. Diabetes Control in Young and Older T2DM Patients

In our study, the mean HbA1c of all patients was 7.8% (±1.4%), with 69.9% of patients exhibiting poor glycemic control (HbA1c > 7%) and 95.9% exhibiting poor diabetes control. Only 4.1% of patients presented favorable diabetes control. In a national survey of 5599 patients with diabetes in Taiwan, 4.1% achieved the ABC standard in 2006 and 8.6% achieved the standard in 2011 [16]. Compared with the achievement rate of 14% demonstrated in the United States from 2007 to 2010 by the United States National Health and Nutrition Examination Survey [3], the number of patients achieving the ABC standard in Taiwan is low. Managing diabetes has become a considerable challenge, especially for individuals 65 years and younger. Literature also showed that younger diabetics had poor glycemic control [17] and increased risks in mortality and complications [2] and need more consideration. 

In this study, the younger patients exhibited poorer glycemic control than did the older patients, especially regarding the control of blood pressure and LDL-C. This may be due to the higher proportion of overweight and obese people among the younger patients with diabetes (≤65 years) (71.9%) compared with that of the older patients (>65 years) (54.1%). Obesity increases insulin resistance, which in turn results in poor glycemic control. This phenomenon emphasizes the challenge of controlling obesity, particularly in men aged ≤65 years.

Moreover, number of years with diabetic diagnose was a significant factor for glycemic control in this study. The risk of poor glycemic control kept increased by 6.7%~7% in every year with diabetes diagnosed (Table 4). This result indicated the importance of number of years with diabetes diagnosis. Therefore, preventing metabolic syndrome and delaying the occurrence of diabetes may lower the risk of hyperglycemia, as well as the occurrence of complications, especial for younger adults since they are expected to have longer life than the older adults. 

### 4.2. Habitual Physical Activity in Young and Older T2DM Patients

The results of this study showed that the duration of moderate-intensity activity minutes a day was significantly associated with normal glycemic control, after adjusting for age, sex, and number of years diagnosed (Table 4). Every minute a day increased in performing moderate-intensity activity lowered the risk of poor glycemic control by 0.1%. This result was compatible with previous studies showing that physical activity significantly reduced HbA1c levels by 0.43~0.67% [6,7,8]. Performing moderate-intensity physical activity every day is important for glycemic control, even though daily physical activity differs from exercise [11], with daily physical activity broadly covering exercise, work, walking, housework, gardening, leisure activity, and transportation. The results of this study suggest that any kind of moderate-intensity physical activity counts and not only exercise effects diabetes control. Moderate-intensity activities can be and should be implemented in a person’s daily schedule. Since the risk of poor glycemic control was increased by 6.7% in every number of years diagnosed, but lowered by 0.1% in every minute a day in performing moderate-intensity activity in this study, performing 67 min of moderate physical activity per day is required to reverse the risk of poor glycemic control by time. In this study, the older patients with diabetes spent 141.7 min per week performing moderate- to high-intensity activity, whereas the younger patients undertook 114.2 min per week. The amount of time engaged in moderate- to high-intensity physical activity was less than that recommended by the ADA, which is 150 min per week [3]. Although physical activities include exercise, leisure activities, or house activities, patients in this study did not achieve this goal of 150 min per week. Only one-fourth to one-fifth of patients engaged in moderate- to high-intensity activity at the recommended frequency of a minimum of 3 days per week. These results are similar to those of Wen et al. (2011), in that East Asians, such as Taiwanese people, undertook physical activity less frequently and tended to undertake low-intensity activity, with only one-fifth of East Asians achieving the suggested goals [18].

In addition, the total amount of physical activity engaged in by the older patients with diabetes in the present study was higher than that of the younger patients, which may be due to 88.3% of the older patients being retired or unemployed. Younger patients generally work and may not have sufficient time to undertake daily physical activities or may not pay too much attention to these physical activities and skip them.

### 4.3. Limitations and Implications 

There are some limitations in this study. The participants were recruited using a non-probabilistic convenience sampling process, which may limit the generalizability of the findings. In addition, a short version of the IPAQ was used to quantitatively measure daily physical activity in this study. This measure is subjective, with recall bias possibly causing underestimation or overestimation. However, this seven-item instrument possesses a certain degree of reliability and validity and can achieve a level of physical activity assessment rapidly [13]. This measure is a time- and labor-saving method. In the future, objective instruments, such as an accelerator, may facilitate the investigation of empirical studies on precisely assessing daily physical activity levels. It may also act as a “feedback” to remind people to engage in physical activities. Moreover, physical activity can be divided into three types: home activity, occupational activity, and leisure activity [19]; therefore, there are many opportunities to perform these activities. Future studies that investigate people’s difficulties and motivations regarding physical activity thoroughly and identify and overcome factors that affect undertaking physical activity are needed. Whether gender and various types of physical activity undertaken by patients with T2DM exhibit an interaction effect on diabetes control must be studied as well. Thus, the goal of controlling diabetes through physical activities can be achieved.

## 5. Conclusions and Recommendations

In clinical care, patients with diabetes cannot easily meet the criteria of the ABC standard. Few studies in Taiwan or in other Asian countries have reported the effects of daily physical activity on diabetes control. The present study found that undertaking moderate daily physical activities was significantly associated with better glycemic control. Changes in lifestyle are extremely critical, and non-drug treatment involving physical activity is essential. Younger diabetics is a group of people that needs further consideration. We suggest that patients implement varied physical activities into their daily life and engage in more moderate physical activities for controlling diabetes. Health providers that work with physical exercise professionals may improve the care in performing physical activities for diabetic patients. Future studies to develop a suitable and workable physical activity program or regimen and to investigate people’s factors for undertaking physical activities may improve the percentage of those undertaking physical activities.

## Figures and Tables

**Figure 1 ijerph-18-01330-f001:**
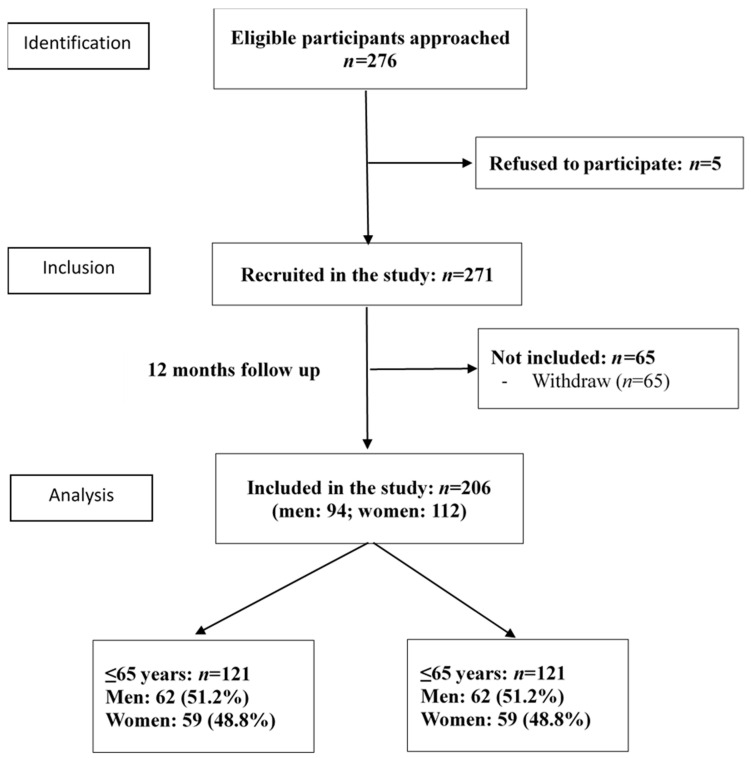
Recruiting flow chart. A total of 216 patients were recruited from an endocrinology outpatient clinic of a medical center. Eleven patients with missing data were excluded from this study. A total of 206 patients completed the baseline and 12-month follow-up measures. Among them, 121 younger patients (age ≤65 years) and 85 older patients (age >65 years) were analyzed separately in this study.

**Table 1 ijerph-18-01330-t001:** Demographic Characteristics and Status of Diabetes Control in Younger (≤65 Years) and Older (>65 Years) Patients with T2DM.

Variable	All Patients		≤65 Years		>65 Years			
	*n*	%	*n*	%	*n*	%	X^2^	*p*
Gender	206	100	121	58.7	85	41.3	3.718	0.054
Men	94	45.6	62	51.2	32	37.6		
Women	112	54.4	59	48.8	53	62.4		
Occupation							**34.191**	**0.000**
Unemployed/retired	138	67	63	52.1	75	88.3		
Working	68	33	58	47.9	10	11.8		
Education level							**20.265**	**0.000**
Illiterate	25	12.1	5	4.1	20	23.5		
<12years	147	71.4	90	74.4	57	67.1		
>12years	34	16.5	26	21.5	8	9.4		
Marital status							2.910	0.088
Married	160	77.7	99	81.8	61	71.8		
Single/Divorced/separated/Widowed	46	22.3	22	18.2	24	28.2		
Number of years diagnosed							**26.732**	**0.000**
1 to 5 years	27	13.1	23	19	4	4.7		
6 to 10 years	58	28.2	40	33.1	18	21.2		
11 to 15 years	55	26.7	32	26.4	23	27.1		
16 to 20 years	35	17	19	15.7	16	18.8		
>20 years	28	13.6	6	5	22	25.9		
Primary caregiver							**7.896**	**0.005**
Husband or wife	126	61.2	84	69.4	42	49.4		
Others	79	38.5	37	30.6	42	50.0		
Living condition								
Living with spouse or children	187	91	110	90.9	77	90.6	2.954	0.228
Solitary	13	6.3	6	5	7	8.2		
Others	3	1.5	3	2.5	0	0		
BMI, kg/m^2^							**14.26**	**0.003**
Underweight (<18.5)	4	1.9	3	2.5	1	1.2		
Normal(18.5~24)	69	33.5	31	25.6	38	44.7		
Overweight(24–27)	61	29.6	33	27.3	28	32.9		
Obese(>27)	72	35	54	44.6	18	21.2		
Medications								
Metformin or other oral drugs	163	79.1%	98	81.0%	65	76.5%	1.046	0.790
Insulin injection	30	14.6%	17	14.0%	13	15.3%		
Combine insulin and other oral drugs	11	5.3%	5	4.1%	6	7.1%		
Glycemic control							0.556	0.456
HbA1c ≤ 7%	62	30.1	34	28.1	28	32.9		
HbA1c > 7%	144	69.9	87	71.9	57	67.1		
Blood pressure control ^1^							**13.75**	**0.000**
Normal	76	36.9	32	26.4	44	51.8		
Abnormal	130	63.1	89	73.6	41	48.2		
LDL-C control							**6.541**	**0.011**
≤100 mg/dl	87	42.2	42	34.7	45	52.9		
>100 mg/dl	110	53.4	73	60.3	37	43.5		
Diabetes control ^2^							3.823	0.051
Normal	8	3.9	2	1.7	6	7.1		
Abnormal	189	91.7	113	93.4	76	89.4		
Complications ^3^								
Neuropathy	34	16.5	21	17.4	13	15.3	0.154	0.695
Cardiovascular	36	17.5	14	11.6	22	25.9	**7.091**	**0.008**
Nephropathy	13	6.3	11	9.1	2	2.4	3.834	0.050
Eye problems	59	28.6	14	11.6	45	52.9	**41.811**	**0.000**

Note: ^1^. Normal blood-pressure control: systolic blood pressure < 130 mmHg and diastolic blood pressure < 80 mmHg for aged ≤65 years; systolic blood pressure < 140 mmHg and diastolic blood pressure < 90 mmHg for aged >65 years. ^2^. Overall diabetes control: Normal = all of the HbA1c, blood pressure, and LDL-C are within normal range; abnormal: one of the HbA1c, blood pressure, and LDL-C are out of normal range. ^3^. Neuropathy includes trembling feet and diabetes foot. Vascular disease includes hypertension, coronary artery syndrome and stroke. Nephropathy is defined as urine albumin-to-creatinine ratio ≥ 30 mg albumin/g creatinine, microalbuminuria ≥ 30 mg albumin/g creatinine, and macroalbuminuria ≥ 300 mg albumin/g creatinine. Eye problems includes retinopathy, cataracts, and glaucoma. Because of missing values, the numbers for some variables do not add up to 206. Statistic significant is indicated with bold.

**Table 2 ijerph-18-01330-t002:** Status of Daily Physical Activities in Younger (≤65 Years) and Older (>65 Years) Patients with T2DM.

	All Patients		≤65 Years		>65 Years			
	Mean	SD	Mean	SD	Mean	SD	t	*p*
Frequency of performing physical activity (day/week)								
High intensity physical activities	0.36	1.316	0.5	1.6	0.1	0.7	**2.248**	**0.026**
Moderate intensity physical activities	1.22	2.463	0.9	2.0	1.7	2.9	**−2.32**	**0.022**
Low intensity physical activities	4.02	2.927	3.8	2.8	4.3	3.1	−1.325	0.187
Amount of time spent on physical activities (minute/day)								
High intensity physical activities	11.54	46.953	13.2	47.4	9.2	46.5	0.605	0.546
Moderate intensity physical activities	15.78	38.615	13.8	41.0	18.6	34.9	−0.894	0.373
Low intensity physical activities	43.77	66.508	47.1	75.4	39.1	51.6	0.855	0.393
Sedentary activities ^1^	300.23	131.759	290.7	133.7	314.3	128.5	−1.158	0.249
Amount of time spent on physical activities (minute/week) ^2^								
High intensity physical activities	40.6	202.1	51.8	226.8	24.7	160.6	0.946	0.345
Moderate intensity physical activities	85.0	232.8	62.4	234.9	117.0	227.2	−1.673	0.096
Low intensity physical activities	254.0	448.3	260.6	504.0	244.7	357.0	0.25	0.803
Total physical activities (METs)	1388.0	1954.5	1373.4	2183.0	1408.9	1585.5	0.946	0.345
	*n*	%	*n*	%	*n*	%	X^2^	*p*
Intensity of physical activities							0.729	0.695
Performing high intensity of physical activities ^3^	24	11.7	13	10.7	11	12.9		
Performing moderate intensity of physical activities ^4^	105	51.0	60	49.6	45	52.9		
Performing low intensity of physical activities ^5^	77	37.4	48	39.7	29	34.1		

Note: ^1^. Sedentary behaviors: including sitting for working, reading, watching TV, eating, talking, etc. ^2^. Amount of time per day) × (Frequency of performing physical activities per week). ^3^. High-intensity physical activities: Total physical activity > 3000 METs or high-intensity activity > 1500 METs a week; ^4^. Moderate-intensity activity: Total physical activity > 600 METs or 20 min of high-intensity activity, 30 min of moderate- to high-intensity activity for 5 days per week, or 30 min of walking for 7 days per week; ^5^. Low-intensity activity: does not meet the aforementioned criteria. Statistically significant is indicated with bold.

**Table 3 ijerph-18-01330-t003:** Association of Performing Physical Activity with Diabetes Control in Patients with T2DM.

	Glycemic Control		Blood Pressure Control ^3^		LDL-C Control		Diabetes Control ^4^	
	r	*p*	r	*p*	r	*p*	r	*p*
All patients								
Frequency of performing physical activity (day/week)								
High intensity physical activities	−0.040	0.564	**0.155**	**0.027**	0.102	0.155	0.036	0.612
Moderate intensity physical activities	**−0.157**	**0.024**	−0.092	0.190	−0.063	0.378	−0.022	0.761
Low intensity physical activities	0.004	0.950	−0.126	0.071	−0.026	0.717	−0.088	0.217
Amount of time spent on physical activities (minute/day)								
High intensity physical activities	0.013	0.849	0.076	0.280	−0.018	0.801	−0.048	0.502
Moderate intensity physical activities	−0.113	0.105	−0.002	0.982	0.019	0.788	0.010	0.894
Low intensity physical activities	−0.049	0.485	−0.003	0.971	0.015	0.832	0.038	0.602
Sedentary activities ^1^	0.042	0.584	−0.12	0.115	−0.044	0.578	−0.094	0.232
Amount of time spent on physical activities (minute/week) ^2^								
High intensity physical activities	0.054	0.439	0.123	0.077	0.027	0.704	0.018	0.805
Moderate intensity physical activities	**−0.163**	**0.019**	−0.056	0.422	0.033	0.642	−0.003	0.962
Low intensity physical activities	−0.066	0.348	−0.015	0.826	−0.008	0.906	0.025	0.724
Total physical activities (METs)	−0.099	0.158	0.036	0.610	0.029	0.687	0.027	0.711
Intensity of physical activities ^5^	−0.113	0.106	−0.009	0.903	0.020	0.783	−0.036	0.613
≤65 years								
Frequency of performing physical activity (day/week)								
High intensity physical activities	−0.088	0.338	0.135	0.141	0.116	0.218	0.050	0.597
Moderate intensity physical activities	**−0.206**	**0.024**	0.054	0.558	−0.076	0.419	−0.067	0.476
Low intensity physical activities	0.006	0.944	−0.131	0.151	0.003	0.972	0.003	0.974
Amount of time spent on physical activities (minute/day)								
High intensity physical activities	0.048	0.604	0.043	0.641	0.069	0.465	0.014	0.881
Moderate intensity physical activities	−0.107	0.244	0.106	0.249	0.008	0.934	0.025	0.790
Low intensity physical activities	−0.007	0.937	−0.136	0.140	0.008	0.934	0.071	0.454
Sedentary activities ^1^	0.046	0.646	−0.040	0.688	−0.031	0.765	−0.095	0.354
Amount of time spent on physical activities (minute/week) ^2^								
High intensity physical activities	0.049	0.591	0.104	0.257	0.077	0.413	0.041	0.66
Moderate intensity physical activities	−0.162	0.076	0.066	0.470	0.048	0.609	0.020	0.830
Low intensity physical activities	−0.021	0.815	−0.132	0.149	−0.007	0.939	0.070	0.458
Total physical activities (METs)	−0.063	0.495	0.005	0.953	0.066	0.483	0.084	0.372
Intensity of physical activities ^5^	−0.080	0.381	−0.036	0.693	0.048	0.614	0.023	0.803
>65 years								
Frequency of performing physical activity (day/week)								
High intensity physical activities	0.067	0.546	0.139	0.207	−0.011	0.919	−0.007	0.949
Moderate intensity physical activities	−0.103	0.35	−0.141	0.197	0.002	0.983	−0.030	0.791
Low intensity physical activities	0.013	0.908	−0.077	0.484	−0.021	0.851	−0.126	0.258
Amount of time spent on physical activities (minute/day)								
High intensity physical activities	−0.04	0.719	0.099	0.369	−0.152	0.173	−0.122	0.274
Moderate intensity physical activities	−0.117	0.285	−0.118	0.281	0.064	0.565	−0.005	0.967
Low intensity physical activities	−0.145	0.187	0.190	0.082	0.000	0.997	0.037	0.738
Sedentary activities ^1^	0.047	0.700	−0.188	0.119	−0.038	0.762	−0.148	0.233
Amount of time spent on physical activities (minute/week) ^2^								
High intensity physical activities	0.057	0.606	0.129	0.238	−0.098	0.383	−0.008	0.946
Moderate intensity physical activities	−0.153	0.163	−0.150	0.171	0.062	0.581	−0.006	0.959
Low intensity physical activities	−0.157	0.151	0.176	0.107	−0.022	0.842	0.016	0.890
Total physical activities (METs)	−0.170	0.119	0.101	0.358	−0.035	0.753	0.002	0.985
Intensity of physical activities ^5^	−0.151	0.168	0.061	0.581	0.008	0.941	−0.084	0.453

Note: ^1^. Sedentary behaviors: including sitting for working, reading, watching TV, eating, talking, etc. ^2^. (Amount of time per day) × (Frequency of performing physical activities per week). ^3^. Normal blood-pressure control: systolic blood pressure <130 mmHg and diastolic blood pressure <80 mmHg for aged ≤65 years; systolic blood pressure < 140 mmHg and diastolic blood pressure < 90 mmHg for aged >65 years. ^4^. Overall diabetes control: Normal = all of the HbA1c, blood pressure, and LDL-C are within normal range; abnormal: one of the HbA1c, blood pressure, and LDL-C are out of normal range. ^5^. Intensity of physical activities: Low-, Moderate-, and High-intensity activity. Statistically significant is indicated with bold.

**Table 4 ijerph-18-01330-t004:** Multiple logistic regression analysis of the association of physical activity with diabetes control in patients with T2DM.

	Univariate					Multiple				
	Wald	*p*	OR	95% CI		Wald	*p*	OR	95% CI	
On Glycemic Control (HbA1c)										
Frequency of performing moderate physical activity (day/week)	**4.909**	**0.027**	**0.879**	**0.784**	**0.985**	3.571	0.059	0.889	0.787	1.004
Age (≤65 vs. >65) ^1^	0.555	0.456	0.796	0.436	1.452	1.557	0.212	0.648	0.328	1.281
Sex (female vs. male) ^2^	0.047	0.829	0.936	0.515	1.701	0.212	0.645	0.863	0.462	1.614
Number of years diagnosed	**4.925**	**0.026**	**1.054**	**1.006**	**1.104**	**6.467**	**0.011**	**1.067**	**1.015**	**1.121**
Performing moderate intensity physical activities (minutes/week)	**4.199**	**0.040**	**0.998**	**0.997**	**1.000**	**3.864**	**0.049**	**0.999**	**0.997**	**1.000**
Age (≤65 vs. >65) ^1^	0.555	0.456	0.796	0.436	1.452	1.813	0.178	0.626	0.316	1.238
Sex (female vs. male) ^2^	0.047	0.829	0.936	0.515	1.701	0.289	0.591	0.842	0.449	1.578
Number of years diagnosed	**4.925**	**0.026**	**1.054**	**1.006**	**1.104**	**6.943**	**0.008**	**1.070**	**1.017**	**1.124**
On Blood Pressure										
Frequency of performing high physical activity (day/week)	3.465	0.063	1.513	0.978	2.341	3.130	0.077	1.495	0.958	2.335
Age (≤65 vs. >65) ^1^	**13.345**	**0.000**	**0.335**	**0.186**	**0.602**	**9.556**	**0.002**	**0.364**	**0.192**	**0.691**
Sex (female vs. male) ^2^	0.009	0.926	0.973	0.552	1.718	0.912	0.339	0.741	0.401	1.371
Number of years diagnosed	**4.447**	**0.035**	**0.961**	**0.925**	**0.997**	0.391	0.532	0.987	0.948	1.028

Note. ^1^. Age ≤65 is the reference. ^2^. Female is the reference. Statistically significant is indicated with bold.

## Data Availability

Not applicable because it does not contain any individual persons’ data. Please contact the corresponding author for data requests.

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
