# Peer review of "Habitual Physical Activity and Diabetes Control in Young and Older Adults with Type II Diabetes: A Longitudinal Correlational Study"

_ijerph, 2021, doi:10.3390/ijerph18031330_

Round 1

Reviewer 1 Report

The gap of the manuscript entitled “Habitual Physical Activity and Diabetes Control in Young and Older Adults with Type II Diabetes: A Longitudinal Correlational Study” is explicitly identified.

The manuscript has potential. However, I recommend authors to make some changes on the structure and format of the manuscript.

Firstly, in the Introduction section in the sentence “Numerous studies have shown that exercise undertaken by patients with T2DM facilitates disease control”, it is necessary to include references that support this sentence.

Secondly, in Methods section, the format of Figure 1 should improve. In addition, authors should describe more specifically the details of the data analysis subsection.

Thirdly, in Discussion section, particularly in Habitual Physical Activity in Young and Older T2DM Patients subsection, it is necessary to compare the results with those obtained by previous literature on the field.

In addition, the limitations of the study should be included as well as some reference to the Informed Consent Statement. Furthermore, references included in the reference list should be carefully checked and adapted to the format required by the journal.

Finally, I congratulate the authors for their research.

Reviewer 2 Report

Dear Authors,

Before indicating the corrections, I thank you for the effort and work. I then give you some reflections, by which I determine "Accept after minor revision".

Introduction

I think that a greater effort should be made to introduce and discuss how the years of diagnosis have a modulating effect. In table 4, it is shown that the years of diagnosis are a data to be taken into account being in all cases significant. 

Since the time of diagnosis is a differentiating aspect, it is necessary to indicate its importance, at least in the discussion. Furthermore, it is necessary to recognise the importance of this and its predictive value in terms of physical activity.

Materials and Methods

In my opinion, the weakness of work is not having stratified the ages studied. It is not the same age range of 32 to 64 as 65 to 82 and indicate the number of women and men in each.

Please, you use "≤65 years" instead of "<= 65 years".

Results

In the notes of the tables, separate them in this way to improve their display, for example: "1.-"

In tables 3 and 4, indicate the data that have statistical significance. Those that are statistically significant and those that are not are not identified. 

In addition, table 3 clearly indicates the three subcategories shown: all patient; ≤65 years and >65 years. Also, in table 4, display the differences between the analyses for "glycemic control" data and those for "blood pressure". Finally, also in table 4, include Performing moderate intensity physical activities (minutes/week) in "blood pressure", as has been done in "glycemic control".

I would invited for authors to use the Odds Ratio (OR). This is a commonly used effect measure for communicating the results of health research. Mathematically, an OR corresponds to a ratio between two odds (e.g., ≤65 years and >65 years), with an odds being an alternative way of expressing the possibility of the occurrence of an event of interest or the presence of an exposure.

Conclusion

I would ask and invite the authors, given the conclusions of the study, to cite the importance of working together with physical exercise professionals to improve the pathology studied. 

References

It is necessary to review all references and update them to the format of the journal. I recommend reviewing the standards in:
https://www.mdpi.com/authors/references 

In addition, review the flaws in some of the references, e.g. Nam et al. (2019). This reference is missing the page (this is 1405). https://www.mdpi.com/2077-0383/8/9/1405 

Finally, once again, I would like to thank the authors for their efforts.

Best regards,

Reviewer 3 Report

The manuscript is very well-written and presented. It can be published as it is interesting for the audience of the journal. 

Minor questions: 

did you considered the international classifications of diseases to considered diabetes in each patient?

What about medications? are they taking metformin or insulin or other anti-diabetic oral drugs? please add this information in your analyses and tables

Any diabetes complications? i.e. retinopathy? 

How many year they have diabetes? add this into your analyses.
